# Tryptophan Metabolism Is Associated with BMI and Adipose Tissue Mass and Linked to Metabolic Disease in Pediatric Obesity

**DOI:** 10.3390/nu14020286

**Published:** 2022-01-11

**Authors:** Julia Lischka, Andrea Schanzer, Margot Baumgartner, Charlotte de Gier, Susanne Greber-Platzer, Maximilian Zeyda

**Affiliations:** Clinical Division of Pediatric Pulmonology, Allergology and Endocrinology, Department of Pediatrics and Adolescent Medicine, Comprehensive Center for Pediatrics, Medical University of Vienna, 1190 Vienna, Austria; julia.lischka@meduniwien.ac.at (J.L.); andrea.schanzer@meduniwien.ac.at (A.S.); margot.baumgartner@meduniwien.ac.at (M.B.); charlotte.degier@meduniwien.ac.at (C.d.G.); susanne.greber-platzer@meduniwien.ac.at (S.G.-P.)

**Keywords:** childhood, aromatic amino acids, inflammation, prediabetes, metabolically healthy obesity, indoleamine 2,3-dioxygenase

## Abstract

The obesity epidemic has contributed to an escalating prevalence of metabolic diseases in children. Overnutrition leads to increased tryptophan uptake and availability. An association between the induction of the tryptophan catabolic pathway via indoleamine 2,3-dioxygenase (IDO) activity and obesity-related inflammation has been observed. This study aimed to investigate the impact of pediatric obesity on tryptophan metabolism and the potential relationship with metabolic disease. In this prospective cohort study, plasma kynurenine, tryptophan, and serotonin levels were measured by ELISA, and IDO activity was estimated by calculating the kynurenine/tryptophan ratio in a clinically characterized population with severe obesity (BMI ≥ 97th percentile) aged 9 to 19 (*n* = 125). IDO activity and its product kynurenine correlated with BMI z-score and body fat mass, whereas concentrations of serotonin, the alternative tryptophan metabolite, negatively correlated with these measures of adiposity. Kynurenine and tryptophan, but not serotonin levels, were associated with disturbed glucose metabolism. Tryptophan concentrations negatively correlated with adiponectin and were significantly higher in prediabetes and metabolically unhealthy obesity. In conclusion, BMI and body fat mass were associated with increased tryptophan catabolism via the kynurenine pathway and decreased serotonin production in children and adolescents with severe obesity. The resulting elevated kynurenine levels may contribute to metabolic disease in obesity.

## 1. Introduction

The dramatic increase in pediatric obesity foreshadows the substantial burden of non-communicable disease that will strain the health care and social systems in the future [1,2,3,4]. Comorbidities associated with obesity include cardiovascular diseases, insulin resistance, type 2 diabetes, hypertension, and dyslipidemia [5,6,7].

Though obesity and the development of non-communicable diseases are closely associated [8,9], a higher BMI is not necessarily indicative of worse metabolic health. The underlying pathomechanisms, particularly considering divergent obese phenotypes, remain enigmatic. Understanding the various mechanisms of how obesity affects metabolic health is essential to ultimately developing effective prevention and treatment strategies, particularly in youth [10,11].

A chronic low-grade inflammatory state is a well-known trait in obesity [12,13] and a common denominator of obesity-related diseases [12,14,15,16]. Additionally, metabolites like amino acids represent factors that interact with pathways involved in metabolic homeostasis [15,16,17]. In this respect, the tryptophan catabolic pathways seem to be of particular relevance since they are regulated by nutritional and inflammatory signals and linked to caloric intake regulation and metabolic control [18,19,20]. The essential amino acid tryptophan cannot be endogenously produced, and is thus provided exclusively through diet. Overnutrition itself leads to increased tryptophan intake and availability [21,22]. Under normal physiological conditions, about 90% of tryptophan is metabolized via the kynurenine pathway in the liver via tryptophan-2,3-dioxygenase; the residual tryptophan is largely used for serotonin synthesis. The kynurenine pathway can alternatively be extrahepatically initiated by indoleamine 2,3-dioxygenase (IDO) expressed in peripheral tissues and induced by hallmarks of obesity such as inflammatory signals (i.e., TNFα, IL-6] as well as oxidative stress [19,23,24,25,26,27,28].

Few reports have explored the relationship of the tryptophan catabolic pathway with metabolic aspects. It is known that key enzymes are expressed also in adipose tissue and are regulated by nutritional signals [18,20,29]. Increased adipose tissue IDO expression and activity were observed in visceral obesity [30,31] and, accordingly, excessive tryptophan catabolism mediated by IDO activity was reported in obesity [30,31,32,33], while a high-fat diet proved to raise kynurenine levels in animal studies [29,34,35]. IDO activity and kynurenine may be directly linked to insulin resistance [36,37,38,39,40,41] and kynurenine itself may trigger obesity [29,35] and associated metabolic disease [35,38,42,43]. Hence, IDO and kynurenine may represent potential treatment targets [20,35].

Adiponectin is an anti-inflammatory and insulin-sensitizing adipokine that is well known to play a central role in glucose and lipid homeostasis, and plasma levels are typically low in obese individuals [44]. However, an association between adiponectin and the tryptophan pathway has not been described yet.

Altogether, the literature suggests a connection between the tryptophan-kynurenine pathway and obesity-associated metabolic disease. However, most of these data derive from experimental research in animals, and little is known about its relevance in pediatric cohorts. Thus, research on the role of the kynurenine-tryptophan pathway in pediatric obesity and metabolic health remains insufficient. It is of particular interest to identify markers and mechanisms that determine the metabolic health state of pediatric patients already suffering from severe obesity who constitute a high-risk population that should be targeted with prevention and treatment strategies. Therefore, the purpose of this explorative study was to assess the role of the tryptophan-kynurenine-pathway in pediatric obesity and to characterize its relationship with obesity-associated metabolic comorbidities within a cohort of pediatric patients with severe obesity.

## 2. Materials and Methods

### 2.1. Patients

Patients attending the outpatient clinic for obesity and lipid metabolic disorders at the Department of Pediatrics and Adolescent Medicine at the Medical University of Vienna with a BMI above the 97th percentile (referred to as “severe obesity” [45] throughout this manuscript) were prospectively enrolled in this explorative study. All patients between nine and 19 years old were eligible for this study. Exclusion criteria were secondary causes of obesity, e.g., endocrine disorders, genetic, syndromic, and drug-induced obesity; drug-associated elevation of liver enzymes and other causes of liver injury (e.g., Wilson’s disease, hepatitis infection). Of 138 eligible patients, 125 were included. Thirteen patients were excluded because of noncompliance with the study protocol.

All study participants underwent a physical examination. Medical history, clinical and laboratory data were collected for all study participants. Tanner stage of puberty was included in routine physician assessment. Anthropometric measures were taken by standardized methods by the same two nurses throughout this study. Body mass index (BMI kg/m^2^) and the respective percentiles were calculated. Serum and plasma samples were obtained in an overnight fasting state and, for non-routine parameters, frozen at −80 °C until analysis. The homeostasis model of insulin resistance (HOMA-IR) was calculated according to Matthews et al.: fasting insulin (µU/mL) × fasting glucose (mg/dL)/405 [46]. Prediabetes was defined as fasting glucose ≥ 100 mg/dL. Metabolically healthy obesity (MHO) was distinguished from metabolically unhealthy obesity (MUO) according to the consensus-based definition by Damanhoury et al.: HDL-C > 40 mg/dL, triglycerides ≤ 150 mg/dL, systolic and diastolic blood pressure ≤ 90th percentile, fasting glucose ≤ 100 mg/dL [10]. Diet was routinely evaluated in the outpatient clinic. All patients had a normal diet without any specific dietary restrictions and did not take any supplements.

### 2.2. Laboratory Parameters

Kynurenine and tryptophan plasma concentrations were measured by ELISA (ImmuSmol, Pessac, France). IDO activity was defined as the kynurenine to tryptophan ratio according to the manufacturer’s instructions.

### 2.3. Statistics

Data are presented as means ± standard deviations (SD) unless otherwise indicated. Continuous variables were assessed by Pearson correlation and Student’s *t*-test, if normally distributed. Parameters with skewed distributions were appropriately log-transformed prior to the analyses; if normal distribution was not achieved, respective non-parametric statistics were used.

A two-sided *p*-value under 0.05 was considered statistically significant. The confidence interval was set at 95%. Since this study is of explorative character, we did not adjust for multiple testing.

All statistical analyses were performed using IBM SPSS Statistics for Windows, version 25 (IBM Corp., Armonk, NY, USA).

## 3. Results

One hundred twenty-five patients with a mean age of 13 ± 3 years were included in the study. Thirteen patients had prediabetes, and 80 were metabolically unhealthy. Detailed characteristics of the study population are shown in Table 1 and details concerning the distribution among MHO criteria are shown in Appendix A.

### 3.1. Relationship between the Tryptophan Pathway and Body Composition/BMI

While tryptophan levels were independent of BMI z-score and body fat percentage, BMI z-score and body fat mass positively correlated with kynurenine concentrations and IDO activity and negatively correlated with serotonin concentrations (Table 2, Figure 1). These correlations remained significant after correction for age and pubertal stage. The inflammatory markers CRP and IL-6 did not correlate with any metabolites of the tryptophan catabolic pathway, while tryptophan negatively correlated with the central adipokine adiponectin (Table 2). This correlation, however, did not remain significant after correction for age and pubertal stage.

To further explore this relationship, metabolite concentrations were evaluated in extreme quartiles of BMI z-score and body fat mass. When comparing the lowest and highest BMI z-score quartile, IDO (*p* < 0.01) was significantly higher and serotonin (*p* = 0.03) lower in the highest BMI z-score quartile. Kynurenine was significantly higher in the highest BMI z-score (*p* = 0.02), and body fat mass quartile (*p* = 0.04) compared to the lowest.

### 3.2. Metabolic Disease and the Tryptophan Catabolic Pathway

Both tryptophan and kynurenine, but not serotonin, correlated with insulin resistance as assessed by HOMA-IR, insulin and c-peptide, and also when adjusting for age and pubertal stage (Table 2). In addition, tryptophan was significantly higher in children with prediabetes (*p* = 0.04) and metabolically unhealthy children (*p* = 0.04) compared to respective controls (Figure 2).

## 4. Discussion

The individual risk for the development of metabolic diseases in obesity remains largely unpredictable. Mechanisms causing obesity-associated diseases include chronic low-grade inflammation and its interactions with metabolic pathways as a central aspect [16,47,48,49]. IDO, the key enzyme of the tryptophan-kynurenine catabolic pathway, is expressed in adipose tissue and is induced by inflammation, a central feature in increased fat mass and a well-known aspect in the pathogenesis of obesity-related diseases [24,33,50]. Thus, tryptophan and its metabolization to serotonin and kynurenine may link obesity, inflammation, and the metabolic state.

Our explorative study shows that within a high-risk population of pediatric patients with severe obesity, BMI z-score and body fat mass are associated with a shift in tryptophan catabolism, leading to increased breakdown towards the kynurenine pathway and decreased catabolism along the serotonin pathway. Elevated tryptophan [33,51], IDO activity [30,31,32,33], and kynurenine [33,36,51,52], as well as lower serotonin levels [50] in obesity have been described before; however, the possible impact of this imbalance on metabolic health remains unclear. In our cohort, higher BMI z-score and body fat mass were positively associated with kynurenine and negatively with serotonin levels, while, notably, tryptophan was not affected despite increased IDO activity. Thus, it appears that in this cohort with severe obesity, tryptophan catabolism, not tryptophan levels themselves, was affected by the extent of obesity.

The shift from serotonin to kynurenine synthesis observed in our study is already well-known as the “serotonin hypothesis,” which proposes this shift as a vicious cycle in the etiology of depression. We now show that this known dysregulation in tryptophan catabolism may also be a factor in metabolic diseases [53,54].

As a result of this cross-sectional analysis, tryptophan concentrations appear to be independent of the grade of adiposity but still correlate with metabolic disease markers like HOMA-IR, insulin, and c-peptide, as did kynurenine concentrations. These correlations are in line with reports on adult populations showing associations of HOMA-IR with kynurenine [36,37,38,39,40,41] and tryptophan [40,41,55]. Additionally, tryptophan was higher in children with prediabetes and metabolically unhealthy children in our cohort. These findings are consistent with results from adults [27,32,56,57,58]. However, one of the few reports that included a pediatric cohort could not confirm the associations of increased kynurenine levels and IDO activity with obesity and metabolic syndrome in juveniles [59]. These conflicting results highlight the need for further investigations, particularly in pediatric cohorts, and evaluations on the impact of age. Reportedly, age has an impact on kynurenine levels [51], which led to the hypothesis that changes in tryptophan breakdown occur over time [18].

The causal relationship of our observations remains under investigation. Nevertheless, a shift towards kynurenine is most likely detrimental, as kynurenine modulates transcription factors that affect BMI regulation and central control of food intake via interactions with *N*-methyl-d-aspartate (NMDA) receptors of glutamate [18]. Furthermore, deleterious effects on metabolic health could also be induced by kynurenine via the aryl hydrocarbon receptor (AHR). Induction of the AHR has been linked to obesity and hyperglycemia in mice. Furthermore, previous studies have proposed a causal connection between IDO activation, which leads to increased kynurenine levels, and obesity-related comorbidities [29,35]. Moreover, the tryptophan-kynurenine pathway via IDO might also play a role in cardiovascular disease, which warrants further research, particularly in cohorts with obesity [1,60,61,62].

On the other hand, lower serotonin levels may exert several beneficial effects on lipid and glucose metabolism [63]. Though we could not observe any correlations of serotonin levels with any metabolic parameter, the observed downregulation of peripheral serotonin concentrations with increased fat mass and BMI may be a regulatory response to increased obesity-induced stress.

Of note, tryptophan levels were independent of BMI in our cohort with severe obesity but associated with worse metabolic health. Elevated tryptophan levels are a known feature in obesity [33,51], but the underlying mechanisms are poorly understood. Overnutrition itself leads to excess tryptophan intake and availability [21,22], which might explain the stable tryptophan plasma levels despite the increased activity of catabolism in our cohort.

One limitation of our study is that we did not systematically assess diet. Further research including controlled diets is warranted to corroborate the findings of our study.

We found a novel association between the tryptophan pathway and adiponectin. This crucial adipokine correlated significantly with tryptophan levels in our cohort. Thus, adiponectin might represent a possible link between adipose tissue inflammation and metabolic changes in obesity, although this correlation was not independent of age and pubertal stage. Previous studies hinted that adiponectin might regulate the kynurenine pathway in mice subjected to a high-fat diet [34], and another linked increased IDO to lower adiponectin levels [64]. Further studies are needed to corroborate these novel findings.

## 5. Conclusions

Tryptophan catabolism is shifted towards the kynurenine pathway and reduced serotonin production with increasing obesity in pediatric patients. IDO activity could be a key factor linking obesity-associated inflammation and metabolic disease. Details of this important association between obesity, inflammation, and other metabolic pathways involved in lipid and glucose metabolism remain to be elucidated.

## Figures and Tables

**Figure 1 nutrients-14-00286-f001:**
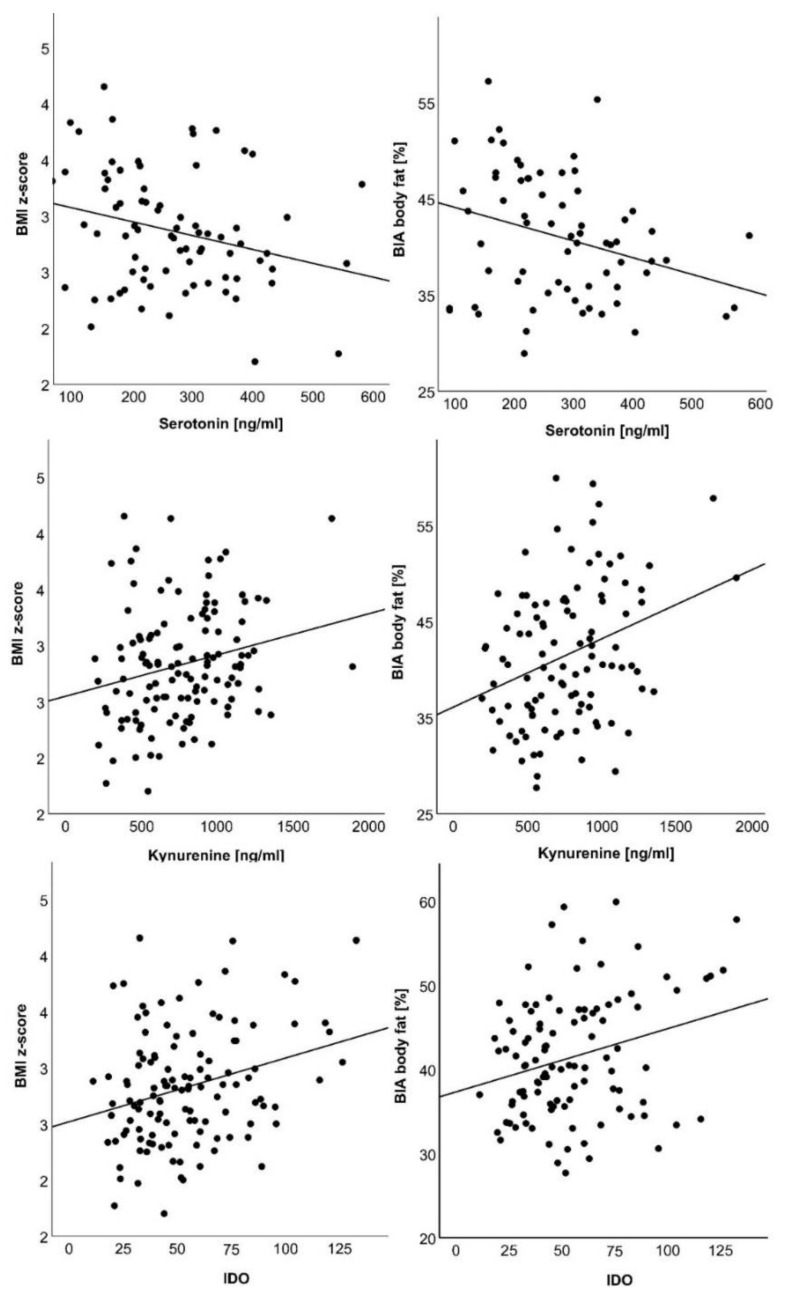
Correlations of IDO activity, kynurenine and serotonin with BMI z-score and body fat percentage (BIA body fat). Respective values are plotted as indicated in the diagram axes.

**Figure 2 nutrients-14-00286-f002:**
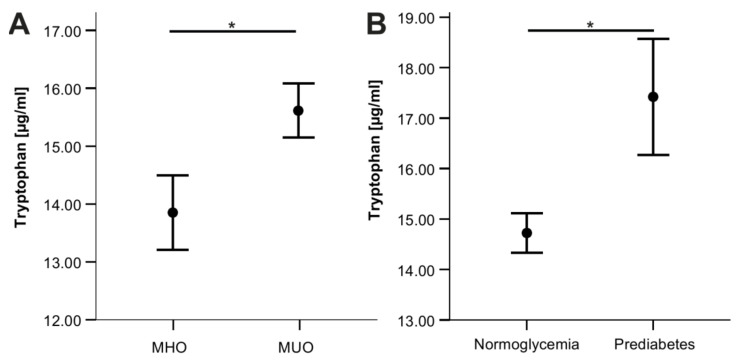
Tryptophan levels of different metabolic conditions. (**A**), metabolically unhealthy (MUO) vs. healthy obesity (MHO; *p* = 0.04). (**B**), prediabetes vs. normoglycemic controls (*p* = 0.04). Diagrams show means ± standard error. * *p* < 0.05.

**Table 1 nutrients-14-00286-t001:** Patient characteristics.

	Mean (*n* = 125)	SD
Sex (*n*)	m:81 f: 44	
Age (years)	13	3
BMI z-score	2.8	0.5
BIA body fat [%]	41.5	7.3
HOMA-IR	6.3	5.20
Insulin [µU/mL]	28.9	20.2
C-Peptide	3.7	2.01
Fasting glucose [mg/dL]	86.0	21.0
Triglycerides [mg/dL]	134.0	90.0
Cholesterol [mg/dL]	168.0	30.0
HDL-C [mg/dL]	43.0	12.0
TNFα [pg/mL]	1.2	0.4
CRP [mg/dL]	0.7	0.6
Adiponectin [µg/mL]	6.5	2.7

**Table 2 nutrients-14-00286-t002:** Correlations of tryptophan, kynurenine, and serotonin with metabolic parameters.

	Tryptophan [µg/mL]	Kynurenine [ng/mL] ^A^	Serotonin [ng/mL]	IDO ^A^
BMI z-score	−0.06	0.23 **	−0.26 **	0.22 *
BIA body fat [%]	0.01	0.29 **	−0.33 **	0.23 *
HOMA-IR ^A^	0.26 **	0.20 *	−0.16	0.02
Insulin [µU/mL] ^A^	0.27 **	0.21 *	−0.15	0.02
C-Peptide ^A^	−0.33 **	−0.26 **	−0.15	0.01
Fasting glucose [mg/dL]	0.02	0.11	−0.16	0.05
Triglycerides [mg/dL] ^A^	0.09	−0.01	−0.13	−0.07
Cholesterol [mg/dL]	0.13	−0.16	0.06	−0.19 *
TNFα [pg/mL]	0.01	0.12	−0.15	0.10
CRP [mg/dL] ^A^	−0.04	0.15	−0.03	0.08
Adiponectin [µg/mL] ^A^	−0.21 *^,B,C^	−0.13	0.15	0.05

^A^ Skewed distribution, thus spearman correlation was calculated; for normally distributed variables, the Pearson correlation was calculated. * *p* < 0.05, ** *p* < 0.01, *p* > 0.05 if adjusted for ^B^ age, ^C^ pubertal stage.

## Data Availability

The data presented in this study are available on request from the corresponding author. The data are not publicly available due to privacy restrictions.

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
