# Peer review of "Tryptophan Metabolism Is Associated with BMI and Adipose Tissue Mass and Linked to Metabolic Disease in Pediatric Obesity"

_nutrients, 2022, doi:10.3390/nu14020286_

Round 1

Reviewer 1 Report

These investigators propose a causative link of tryptophan metabolism to childhood obesity. Their primary theories for a mechanism are either an increase in inflammation associated with the metabolite kynurenine or a metabolic shift away from serotonin production. The study relies on vague correlations with no direct evidence.

There are multiple major flaws in the interpretation of these findings.

  1. The investigators have no information about dietary intake of tryptophan. If Trp is going to be a primary cause of obesity (versus simply a biomarker of a secondary disruption of metabolism) then dietary intake is a critical and essential piece of information.
  2. Investigators have total lack of any metabolic flux data. There is no evidence for any change in flux through the Trp pathways, simply reporting an increase plasma kynurenine concentration may reflect a change in Trp intake, changes in the pathways, or simply a downstream dysregulation associated with calorie imbalance and excessive glucose metabolism.
  3. The data fail to demonstrate any impact of the obesity or Trp on biomarkers of inflammation (CRP, IL-6, or TNF-a).

The entire study is based on weak statistical associations of HOMA and blood insulin with kynurenine. There is no evidence to discern if these associations are primary metabolic disruptions or secondary to abnormal energy balance and carbohydrate metabolism. As with other amino acids, such as the branched-chain amino acids, research has clearly established that the abnormal blood levels are secondary to a disruption in insulin and carbohydrate metabolism. Hence, while elevated kynurenine may be a biomarker for metabolic dysregulation, there are no data provided to suggest dietary Trp or Trp metabolism is a primary cause of obesity.

Author Response

Comments and Suggestions for Authors

These investigators propose a causative link of tryptophan metabolism to childhood obesity. Their primary theories for a mechanism are either an increase in inflammation associated with the metabolite kynurenine or a metabolic shift away from serotonin production. The study relies on vague correlations with no direct evidence.

There are multiple major flaws in the interpretation of these findings.

1. The investigators have no information about dietary intake of tryptophan. If Trp is going to be a primary cause of obesity (versus simply a biomarker of a secondary disruption of metabolism) then dietary intake is a critical and essential piece of information.

Answer: Thank you for the comment. As mentioned in the manuscript (introduction line 60-62, discussion line 175-177), elevated tryptophan and kynurenine and low serotonine levels have been previously described in obesity, and overnutrition itself, underying obesity, leads to increased tryptophan intake and availability (Ref. 21, 22). We assessed diet in the outpatient clinic during the routine visit. All included patients do not have any specific dietary restrictions and thus had a normal  diet. We added this information to the methods section (line 106-107) since we agree this information is of relevance when interpreting our results and additionally added this limitation to the discussion section (line 222-223) However, this assessment was not designed to systematically determine Trp uptake.
Of note, many previous studies investigating tryptophan metabolism did not control for diet, for example studies that also investigated its impact in childhood obesity (Mangge H, et al. Obesity-related dysregulation of the tryptophan-kynurenine metabolism: role of age and parameters of the metabolic syndrome. Obesity (Silver Spring). 2014., Goffredo M et al. A Branched-Chain Amino Acid-Related Metabolic Signature Characterizes Obese Adolescents with Non-Alcoholic Fatty Liver Disease. Nutrients. 2017 Jun 22;9(7):642). 
Nonetheless, we agree that further research including controlled diets is warranted to corroborate the findings of our study and added this notion to the discussion (line 223).

2. Investigators have total lack of any metabolic flux data. There is no evidence for any change in flux through the Trp pathways, simply reporting an increase plasma kynurenine concentration may reflect a change in Trp intake, changes in the pathways, or simply a downstream dysregulation associated with calorie imbalance and excessive glucose metabolism.

Answer: Thank you for this remark. Indeed, in this study we report associations, and we are fully transparent regarding the strength of correlations observed in our analyses, as reported in table 2. We do not claim any causative implications in our results. To emphasize the explorative character of our study we added this information throughout the manuscript (line 119, 172) and also adjusted the title accordingly to reflect that we report associations.

3. The data fail to demonstrate any impact of the obesity or Trp on biomarkers of inflammation (CRP, IL-6, or TNF-a).

Answer: In fact, CRP and IL-6 both correlated significantly with body fat content (R=0.4 and 0.27, respectively) and BMI z-score (R=0.25 and 0.2, respectively) in our cohort, although TNFα did not (not reported in the manuscript). Although we could not show a correlation of CRP, IL-6 and TNFα with parameters of the tryptophan pathway, in order to be transparent, we chose to report the negative results. However, there was a negative association of tryptophan with adiponectin, which is a well-known anti-inflammatory marker. Possibly, an association between inflammatory markers and the tryptophan pathway might be underestimated in our cohort and may become more apparent in larger cohorts with a broader BMI range.

The entire study is based on weak statistical associations of HOMA and blood insulin with kynurenine. There is no evidence to discern if these associations are primary metabolic disruptions or secondary to abnormal energy balance and carbohydrate metabolism. As with other amino acids, such as the branched-chain amino acids, research has clearly established that the abnormal blood levels are secondary to a disruption in insulin and carbohydrate metabolism. Hence, while elevated kynurenine may be a biomarker for metabolic dysregulation, there are no data provided to suggest dietary Trp or Trp metabolism is a primary cause of obesity.

Answer: Thank you for this elaboration. As reported, this pathway has been well-investigated in obesity and mental health (lines 183-185), emphasizing its dysregulation in obesity, but the metabolic implications are not sufficiently investigated yet. Adipose tissue has been mechanistically linked to elevated BCAA levels via downregulation of the key catabolizing enzymes on a transcriptional level  (incl. BCAT and BCKDH) (White PJ, et al. Insulin action, type 2 diabetes, and branched-chain amino acids: A two-way street. Mol Metab. 2021 Oct;52:101261). Hence, the interaction of amino acid metabolism and its relationship to obesity and metabolic disease is complex and poorly understood, and our results connecting dysregulation in tryptophan catabolism to clinical features in our patients may promote research on a basic scientific level to better understand the underlying mechanisms.

We agree that the associations with the HOMA-IR (and insulin) are not very strong but together with the data on MHO and prediabetes our cautious conclusions appear justified.

Of note, in our manuscript we do not claim anywhere that Trp or Trp metabolism may be a primary cause of obesity or that we aimed to prove that hypothesis. We believe, however, that our results may lead to a better understanding of the complex mechanisms leading to metabolic dysregulation in obesity and obesity-associated diseases. 

Reviewer 2 Report

Dear Authors,

Thank you for your work and a very well written manuscript. I have a few thoughts that you may want to consider.

Major:

  1. Title: The word "affected" is strong and may sound like you are proposing causality. I would suggest replacing it with associated.
  2. Manuscript overall- please use people first language and refer to the study population as children with severe obesity and not severely obese children.
  3. Methods: Can you please clarify why you used the BMI criterion of greater than 97th percentile to classify obesity - to my knowledge, this is not the most current and accepted criteria. I have seen the usage of percentage of the 95th percentile BMI for age and sex. It is helpful if we use the same criteria to compare studies in future.
  4. Methods: Diet and exercise can be major confounders which are not assessed/mentioned in the manuscript.
  5. Methods: How was pubertal age/stage assessed?
  6. Methods: Were the participants using any supplements?
  7. Methods: You describe how you categorized MHO but it will be nice to see the breakdown of how many participants met which of the criterion to qualify as MHO?
  8. Results: I am curious about any sexual dimorphism in these results?
  9. Results: Figure 1 shows that there is range of BMI and body fat% at a given tryptophan/kynurenine level. It might be worthwhile to explore the differences in phenotype at different metabolite concentrations (maybe take quartiles and compare extreme quartiles).
  10. Results: Was there any racial diversity in your sample?
  11. Results: Table 2 may be easier to read if you only show the significant associations. I realize that it is an exploratory study and hence controlling for multiple comparisons would affect the power, nonetheless it should be discussed as a limitation so that the readers are clear.
  12. Discussion: Does IDO activity change with age and how will that factor into the analysis?

Thank you again for your work. Best Wishes.

Author Response

Dear Authors,

Thank you for your work and a very well written manuscript. I have a few thoughts that you may want to consider.

Major:

1. Title: The word "affected" is strong and may sound like you are proposing causality. I would suggest replacing it with associated.

Answer: Thank you for your considerate comments. We reworded the title as suggested.

2. Manuscript overall- please use people first language and refer to the study population as children with severe obesity and not severely obese children.

Sorry for this omission. We changed the wording where applicable.

3. Methods: Can you please clarify why you used the BMI criterion of greater than 97th percentile to classify obesity - to my knowledge, this is not the most current and accepted criteria. I have seen the usage of percentage of the 95th percentile BMI for age and sex. It is helpful if we use the same criteria to compare studies in future.

We agree that harmonizing BMI criteria is important in order to compare studies. However, according to our national/regional guidelines, the “Evidence-based Therapy Guideline of the German Working Group on Obesity in Childhood and Adolescence” (https://doi.org/10.1007/s00103-011-1269-2) children and adolescents with a BMI>97th percentile should be referred to thorough clinical examination. Accordingly, children above that cut-off of the 97th percentile are treated at our center and thus could be recruited for this study. In this guideline BMI above 97th percentile is referred to as “obese”. However, using this term is not in line with other international guidelines and would be potentially misleading. This is a known difficulty as, for example, addressed by Flegal and Ogden (Adv Nutr. 2011; 2(2): 159S–166S). Therefore, we stated that we define a BMI above the 97th percentile as “severe obesity” in this manuscript (line 86) and refer to ref 45 as exemplary publication.

4. Methods: Diet and exercise can be major confounders which are not assessed/mentioned in the manuscript.

Thank you for the comment. As mentioned in the manuscript (introduction line 60-62, discussion line 175-177), elevated tryptophan and kynurenine and lower serotonine levels have been previously described in obesity. Exercise was not systemiatically assessed during the routine visit. All included patients do not have any specific dietary restrictions and thus had a normal diet, we added this information to the methods section (line 106-107) since we agree this information is of relevance when interpreting our results and additionally added this limitation to the discussion section (line 222-223). Of note, this study is designated explorative, and further research may aim to confirm our finding also when controlling for exercise and diet.

5. Methods: How was pubertal age/stage assessed?

Pubertal stage was assessed using Tanner stage, which was evaluated during the physical examination by the treating pediatrician (J.L.). We added this information to the methods section (line 94-95).

6. Methods: Were the participants using any supplements?

The participants were not taking any supplements (now mentioned in line 107). Vitamin D and folate are included in our routine blood investigations and were supplemented after blood sampling for the study if low.

7. Methods: You describe how you categorized MHO but it will be nice to see the breakdown of how many participants met which of the criterion to qualify as MHO?

We agree that further information on health status of the study population would help interpreting the results. We added the respective information (Supplementary table 1)

8. Results: I am curious about any sexual dimorphism in these results?

Thank you! We evaluated also influences of sex and observed no significant differences attributable to sexual dimorphisms.

9. Results: Figure 1 shows that there is range of BMI and body fat% at a given tryptop            han/kynurenine level. It might be worthwhile to explore the differences in phenotype at different metabolite concentrations (maybe take quartiles and compare extreme quartiles).

Thank you for this suggestion, we calculated and compared quartiles as suggested and found that indeed when comparing lowest and highest BMI z-score quartile, IDO (p<0.01) was significantly higher and serotonin (p=0.03) lower in the highest BMI z-score quartile. Kynurenine was significantly higher in the highest BMI z-score (p=0.02) and body fat mass quartile (p=0.04) compared to the lowest. We added these findings to the results section (lines 139-143).

10. Results: Was there any racial diversity in your sample?

We assess ethnicity routinely in our outpatient clinic. In this sample, the vast majority was Caucasian, one child was Hispanic and one Asian.

11. Results: Table 2 may be easier to read if you only show the significant associations. I realize that it is an exploratory study and hence controlling for multiple comparisons would affect the power, nonetheless it should be discussed as a limitation so that the readers are clear.

Thank you for the suggestion. We removed the parameters HDL, LDL, IL-6 and ALT from table 2 since we agree they are not of particular relevance. Additionally, we added a statement that we did not control for multiple comparisons to the methods section (line 119-120).

12. Discussion: Does IDO activity change with age and how will that factor into the analysis?

Thank you, we briefly discussed the impact of age on kynurenine, since conflicting results have been reported regarding the impact of age (ref 51, 18; lines 187-199) which may also account in a similar manner for IDO activity as the key enzyme in this pathway. In our cohort, no significant impact of age was detectable, We adjusted analyses for age and found that only the correlation of tryptophan with adiponectin was affected. Also, when looking at IDO activity specifically in association with age, we found a balanced distribution across our age range. Future research may aim to clarify to role of age in tryptophan metabolism and particularly the kynurenine pathway.

Thank you again for your work. Best Wishes.

Thank you so much!

Round 2

Reviewer 2 Report

Thank you for making the edits.